

# Algal growth and weathering crust structure drive variability in Greenland Ice Sheet ice albedo

Andrew J. Tedstone[1], Joseph M. Cook[2], Christopher J. Williamson[1], Stefan Hofer[1],
Jenine McCutcheon[3], Tristram Irvine-Fynn[4], Thomas Gribbin[1], and Martyn Tranter[1]

[1]Bristol Glaciology Centre, School of Geographical Sciences, University of Bristol, Bristol, UK
[2]Department of Geography, University of Sheffield, Sheffield, UK
[3]School of Earth and Environment, University of Leeds, Leeds, UK
[4]Department of Geography and Earth Sciences, Aberystwyth University, Aberystwyth, UK

**Correspondence:** Andrew Tedstone (a.j.tedstone@bristol.ac.uk)

**Abstract.** One of the primary controls upon the melting of the Greenland Ice Sheet (GrIS) is albedo. There is a major difference in the albedo of snow-covered versus bare-ice surfaces, but observations also show that there is substantial spatio-temporal variability of up to ~0.4 in bare-ice albedo. Variability in bare ice albedo has been attributed to a number of processes including the accumulation of Light Absorbing Impurities (LAIs) and the changing physical properties of the near-surface ice. However,
the combined impact of these processes upon albedo remains poorly constrained. Here we use field observations to show that among LAIs, pigmented glacier algae are ubiquitous and cause surface darkening both within and outside the south-west GrIS 'dark zone', but that other factors including modification of underlying ice properties by algal bloom presence, surface topography and weathering crust development are also important in determining patterns of daily albedo variability. We further use unmanned aerial system observations to examine the scale gap in albedo between ground versus remotely-
sensed measurements made by Sentinel-2 (S-2) and MODIS. S-2 observations provide a highly conservative estimate of algal bloom presence because algal blooms occur in patches much smaller than the ground resolution of S-2 data. Nevertheless, the bare-ice albedo distribution at the scale of 20×20 m S-2 pixels is generally unimodal and unskewed. Conversely, bare ice surfaces have a left-skewed albedo distribution at MODIS MOD10A1 scales. Thus, when MOD10A1 observations are used as input to energy balance modelling then meltwater production can be under-estimated by ~2%. Our study highlights that (1)
the impact of physical ice surface processes is of similar importance to the direct darkening role of light-absorbing impurities upon ice albedo and (2) there is a spatial scale dependency in albedo measurement which reduces detection of real changes at coarser resolutions.

## 1   Introduction

The Greenland Ice Sheet (GrIS) has experienced ~2 °C of summer warming since the mid 1990s, increasing runoff by more
than 40 % without concomitant increases in precipitation (van den Broeke et al., 2017). Since approximately 2010 the total mass imbalance has been dominated by melting and runoff, corresponding to 68% of mass losses between 2009 and 2012 (Enderlin et al., 2014). This is especially important on the western side of the ice sheet where the majority of meltwater runs





off directly rather than refreezing (Steger et al., 2017). Enhanced melting has been caused by recent persistent anticylonic summer conditions (Fettweis et al., 2013) which reduce cloud cover, leading to enhanced shortwave radiation over the ablation zone (Hofer et al., 2017). Mass loss from the GrIS accounted for 37% of cryospheric sea level rise from 2012 to 2016 (Bamber et al., 2018) and it is therefore critical to understand the contribution of surface melting and runoff to GrIS mass loss.

Melting is principally controlled by net shortwave radiation which in turn is modulated by surface albedo. Lower albedo snow and ice absorb more energy, leading to faster melting and more runoff. Since around 2000 the albedo in several GrIS sectors has declined, especially along the western margins where albedo reduced by as much as 9% between 2000 and 2017 (van den Broeke et al., 2017). Some of this change can be attributed to winter snowpack melting earlier in the summer, revealing lower albedo ice (Box et al., 2012; Ryan et al., 2019), but observations of surface albedo and reflectance made over the past

$\sim$20 years also show an overall increase in the extent and magnitude of 'dark' ice as distinct from clean bare ice surfaces (Shimada et al., 2016; Tedstone et al., 2017). Albedo is one of the largest uncertainties in energy balance modelling (Hock, 2005; Noël et al., 2015). Models generally fail to capture the magnitude of the albedo reductions which have occurred in 'dark' areas, probably because Light Absorbing Impurities (LAIs) are not presently included in model albedo schemes (Tedesco et al., 2016).

Despite previous studies inferring the potential albedo-reducing importance of impurities including cryoconite, emergent dust and liquid meltwater (Greuell, 2000; Bøggild et al., 2010; Wientjes and Oerlemans, 2010), there is an emerging consensus that pigmented glacier algae grow on the ice surface (Uetake et al., 2010; Yallop et al., 2012; Stibal et al., 2017; Williamson et al., 2018) and are the dominant agent of darkening amongst LAIs (Stibal et al., 2017; Tedstone et al., 2017; Cook et al., 2019b). Glacier algae reduce albedo both directly (i.e. the cells absorb shortwave radiation) and indirectly by modifying the

underlying ice surface, for instance by maintaining a liquid water film (Cook et al., 2017, 2019b; Williamson et al., 2019). They are ubiqitous across south-west Greenland (Cook et al., 2019b; Wang et al., 2018). Their growth is principally controlled by (i) the timing of winter snowpack retreat, (ii) meltwater availability and (iii) sufficient photosynthetically-active radiation (Williamson et al., 2019).

    The physical properties of the uppermost surface ice itself, however, are also important in determining albedo. When short-

wave radiative energy fluxes dominate, a porous, low-density weathering crust develops as a consequence of radiative energy penetration to the sub-surface (Muller and Keeler, 1969; Munro, 1990). This, together with cryoconite hole formation punctuating the porous substrate (McIntyre, 1984; Cook et al., 2016), can allow supraglacially-generated meltwater to drain into a shallow, depth-limited sub-surface water table (Irvine-Fynn et al., 2011; Cooper et al., 2018; Christner et al., 2018). This porous near-surface ice layer typically has numerous air-ice interfaces characterised by a rough surface topography, offering

opportunities for high-angle light scattering, which increases albedo (Jonsell et al., 2003).

    It is difficult to identify the emergent processes that control bare ice albedo over landscape scales because there is a disconnect between the centimetre scales of ground-based spectroscopy versus remote sensing over hundreds of metres by satellite platforms such as MODIS. Ground-based spectroscopy in the south-west 'dark zone' during the 2012 and 2013 seasons showed bare ice albedo variability of 10–30 % and that dirty ice introduced a left-skew in the albedo distribution of transect-based mea-

surements (Moustafa et al., 2015). Single-point-to-satellite-pixel validation is inadequate as there are large in-situ deviations





from coarser-scale satellite albedo measurements, so multiple-point-to-pixel approaches are needed to capture spatial variability (Moustafa et al., 2017; Ryan et al., 2017).

Unmanned aerial systems (UAS) provide one way to bridge the scale gap between ground and satellite observations, by making high spatial resolution measurements over tens of metres to kilometres. This is especially useful for examining hetere-

ogeneous distributions in LAIs. For example, on a single day in 2014, LAIs including dust, black carbon and pigmented algae explained 73 % of spatial variability in albedo along a 25 km transect (Ryan et al., 2018). More recently, combined ground sampling, radiative transfer modelling and surface type classification of UAS and satellite imagery showed that algal blooms specifically can cover at least 78 % of ice in the 'dark zone', generating at least 6–9 % additional ice melt in the south-west 'dark zone' during the dark year of 2016 compared to the 'average' year of 2017 (Cook et al., 2019b). Higher resolution

imagery is therefore able to bridge the scaling gap and has been crucial in demonstrating that glacier algae are the dominant LAI.

Whilst previous studies have made signficant advances in understanding spatial variability in albedo, there remain two key challenges: (1) making measurements elsewhere beyond the 'dark zone', and (2) understanding why surface type and bare ice albedo change through time. Here we present observations of surface type and albedo made by multi-spectral unmanned aerial

system (UAS) paired with ground sampling at two locations along the western GrIS margin. We examine the drivers of the measured albedo patterns, and at one site we also examine changes in albedo through time and undertake a multiple-point-to-pixel comparison to assess whether these changes are captured by the Sentinel-2 and MODIS sensors.

## 2   Study sites

Albedo and surface type measurements were made at two sites in two different years (Fig. 1, inset). During July 2017 we

acquired approximately one week of measurements at S6 (67.07ºN, 49.38ºW, 1073 m asl) located within the south-west 'dark zone' approximately 60 km north-east of Kangerlussuaq and within 2 km of the IMAU S6 automatic weather station (AWS). We also occupied the site from 31 May to 1 July, enabling us to observe the retreat dynamics of the winter snowpack for most of the early ablation season. During June there were several epsiodes of snowpack melting, with most of the snowpack retreating by mid June and exposing bare ice with hetereogeneous albedo. However, a series of large snowfall events occurred

towards the end of June and the ice surface was covered by ∼10 cm snow when we left on 1 July. Most snow had melted away when we re-established the site on 13 July for UAS measurements.

We measured surface type and albedo on a single day, 24 July 2018, at UPE_U (72.88ºN, 53.55ºW, 950 m asl), hereafter UPE. The site was located in the ablation zone, 26 km from the ice margin to the east of Upernavik and ∼670 km north of S6 and was within 2 km of the PROMICE UPE_U AWS. The surface was predominantly bare ice when the field site was

established on 21 July. However, there were then several snowfall events which caused a thin layer of snow to obscure much of the ice surface throughout the campaign. Snow fell on 22, 25, 26 and 27 July. Nevertheless, air temperatures exceeded 0 ºC every day between 21 and 27 July, partially melting the snow between each snowfall event.



## 3 Data and Methods

### 3.1 UAS data

We mapped a 250×250 m area of ice surface at each site using the methodology described previously in Cook et al. (2019b). Briefly, we integrated a MicaSense Red-Edge multispectral camera onto a Steadidrone Mavrik-M quadcopter (referred to

hereafter as UAS). The camera was remotely triggered through the autopilot which was programmed along with the flight coordinates in the open-source software Mission Planner. Images were acquired at approximately 2 cm ground resolution with 60% overlap and 40% sidelap. Mapping required two successive flights with a UAS battery change between them. Each flight lasted ∼ 10 min, was made at 30 m above the ice surface, and took place under clear-sky illumination conditions unless otherwise noted (Appendix A).

At S6 we made UAS flights over several successive days, requiring us to remove the effect of ∼0.5–1 md$^{-1}$ of ice motion from the final orthomosaics. We therefore placed 15 Ground Control Points (GCPs) and measured their X/Y locations on 21 July using a differential Global Navigation Satellite System (GNSS) receiver, post-corrected by reference to the Kellyville International GNSS Service (IGS) GNSS station using IGS final orbits. We used the GCPs to constrain the horizontal geo-referencing of every orthomosaic to the same static georectification solution.

We applied radiometric calibration and geometric distortion correction following MicaSense procedures (MicaSense, 2018). We then converted from radiance to reflectance using time-dependent regression between measurements of the MicaSense Calibrated Reflectance Panel (and, at UPE, a Spectralon® panel) acquired before and after each flight. The individual reflectance-corrected images were mosaiced using AgiSoft PhotoScan following United States Geological Survey (2017), yielding multi-spectral orthomosaics with 5 cm ground resolution. Finally, the orthomosaics were radiometrically adjusted to match directional

reflectance measurements made by ground spectroscopy so that our surface classifier (Sect. 3.4), which was trained using the directional reflectance measurements, could be applied to the orthomosaics.

The orthomosaics were used in three ways: (i) converted to albedo using a narrowband-to-broadband approximation (Knap et al., 1999), (ii) classified into surface types (see Sect. 3.4), and (iii) digital elevation models derived photogrametrically in Agisoft PhotoScan at 5 cm ground resolution.

We used the photogrammetric DEMs to derive (i) study area slope angle and (ii) local topographic variability. To calculate the slope angle we applied a gaussian filter with a window of 0.25 m to remove very-high-frequency topographic features, then we calculated the average slope across each study area after Horn (1981) as implemented in the RichDEM library (Barnes, 2016). To examine local topographic variability ('roughness') we applied a gaussian filter with a window of 4.95 m, then subtracted it from the DEM to yield a detrended surface.

### 3.2 Biological sampling

We took samples of the ice surface at each site to quantify the presence of ice algal cells. At S6, samples were made immediately after collection of paired ground spectra (Sect. 3.4) to enable direct upscaling by UAS imagery analysis. At UPE, widespread snow cover prevented us from utilising the paired approach carried out at S6. Instead, on 26 July (two days after the UAS





flight) we cast a random 75-point sampling grid over our UAS flight area. We used a trowel to scrape the snow away to reveal the bare ice surface beneath for sampling.

Samples were made by cutting a $30{\times}30{\times}2$ cm volume out using a metal ice saw and trowel and transferring into a sterile Whirl-Pak bag which was immediately placed in the dark to melt over a $\sim$24 h period at ambient air temperature. Following

melting, samples were homogenised, sub-sampled into Falcon tubes and fixed with 2% final concentration gluteraldehyde. Samples were then returned to laboratories at the Universities of Sheffield and Bristol for counting by microscopic haeomcy-ometry. Full details of the enumeration protocols used are in Cook et al. (2019b) (samples from 2017) and Williamson et al. (2018) (samples from 2018).

### 3.3 Sentinel-2 data

Clear-sky Sentinel-2 (hereafter S-2) data were available at the S6 site for 20 and 21 July. No clear-sky acquisitions were available coincident with our field season at the UPE site. We downloaded S-2 L1C data from SentinelHub (Sinergise, Slovenia). We used all bands available at 10 and 20 m resolution by resampling those bands delivered at 10 m resolution to 20 m using the S-2 toolbox of the European Space Agency (ESA) 'SNAP' platform. We processed the L1C data to L2A surface reflectance using the ESA Sen2Cor processor. The data were then (i) converted to broadband albedo using a narrowband-to-broadband

approximation (Liang, 2001) and (ii) classified into surface types (see Sect. 3.4).

### 3.4 Surface type classification

To classify images by surface type we used a supervised classification approach following Cook et al. (2017), trained on ground spectra collected at S6 with a FieldSpec Pro 3 (Analytical Spectral Devices, Boulder, USA) during the 2016 and 2017 field seasons at S6. Briefly, we used 171 directional reflectance measurements. The measurements were labelled by visual

examination as snow ('SN'), water ('WA'), clean ice ('CI'), light algae ('LA'), heavy algae ('HA') and dispersed cryoconite ('CC'). After ground spectra were acquired we took destructive ground samples following procedures in Sect. 3.2. Clean ice samples contained $625 \pm 381$ cells ml$^{-1}$, light algae samples $4.73{\times}10^3 \pm 2.57{\times}10^3$ cells ml$^{-1}$ and heavy algae samples $2.9{\times}10^4 \pm 2.01{\times}10^4$ cells ml$^{-1}$, confirming the accuracy of our visual assessments of each surface type. We split the dataset randomly into training (70%) and test (30%) sets. These data were used to train a Random Forest classifier, which had the

highest performance of all classifiers tested (Cook et al., 2019b). We trained the algorithm to predict surface type from (i) our UAS-acquired data, utilising all 5 bands of data, and (ii) S-2 data, utilising all 9 bands at 20 m resolution. The confusion matrices (Appendix B) for the classifiers in this study were similar to those in Cook et al. (2019b). Against the test set, UAS classifier accuracy and recall were both 97% and S-2 classifier accuracy and recall were both 88%.

### 3.5 MOD10A1 data

We used the albedo retrievals contained within the MODIS/Terra Snow Cover Daily L3 Global 500 m Grid V006 'MOD10A1' data product (Hall and Riggs, 2016). The two pixels which overlapped with our S6 UAS area were examined in their origi-



nal sinusoidal projection. Precise overpass times were extracted from the granule pointer information contained within each product file (Appendix A). There were no cloud-free MOD10A1 data available at UPE during our field season.

## 3.6 Energy balance and melt modelling

To provide a local environmental context we used a point surface energy balance model (Brock and Arnold, 2000) to estimate net shortwave and longwave radiation fluxes, the turbulent sensible and latent heat fluxes and the surface melt rate at a point on a melting ice or snow surface. The model was forced at an hourly timestep by continuous measurements of shortwave radiation, vapour pressure, air temperature and wind speed made by IMAU S6 AWS (Kuipers Munneke et al., 2018) and PROMICE UPE_U AWS (van As et al., 2011). We used the albedo measured at each AWS, which at UPE_U was only for solar zenith angles below 70$^\text{o}$ and at S6 was only when downwelling shortwave radiation was >250 W m$^{-2}$; night-time values were therefore forward-filled from the last valid albedo observation. The surface roughness length was held constant at 1 mm according to similar values for ablating ice surfaces (Brock and Arnold, 2000). As the AWS were located a few km away the computed melt rates should be interpreted as indicative of the meteorologically-forced melting regime rather than as absolute melt rates experienced across the study areas.

## 4 Results and Discussion

### 4.1 Impact of glacier algae

Glacier algae were ubiquitous at both S6 and UPE. At S6, low albedo (Fig. 1a) was caused by extensive algal blooming (Fig. 1b) enabled by melting over several preceding weeks (see Cook et al., 2019b). This finding is supported by radiative transfer modelling which shows that mineral dusts local to S6 are weakly absorbing and strongly scattering, meaning that they locally increase albedo, whereas glacier algae have an albedo-reducing effect (Cook et al., 2019b). At UPE, the albedo was higher (Fig. 1d) due to persistent snow cover obsuring the darker bare ice surface (Fig. 1e). However, our ground sampling revealed up to 80% LA+HA coverage of the survey area (Appendix C) on the bare ice surface that was hidden from our aerial remote sensing by a layer of fresh snow. Ultrasonic ranging observations from the UPE_U AWS show that the winter snowpack had melted by 29 June 2018, revealing the bare ice beneath. Between bare ice exposure and our arrival at the field site the surface had remained snow-free and our energy balance modelling estimates that 35 cm w.e. of melt had occurred. These conditions promote algal growth (Yallop et al., 2012; Williamson et al., 2018; Stibal et al., 2017), explaining the presence of algae beneath the recently-deposited snow. These observations of spatially expansive populations of algae at both sites demonstrate that biological albedo reduction is important across the ablation zone of the western GrIS including areas outside of the 'dark zone'.

Albedo was a weak predictor of surface class, with considerable overlap in the albedo of the various classes (Fig. 1c,f). Broadband albedo alone is therefore not a reliable predictor of ice surface type and cannot be used to infer the presence of glacier algae or other LAIs.
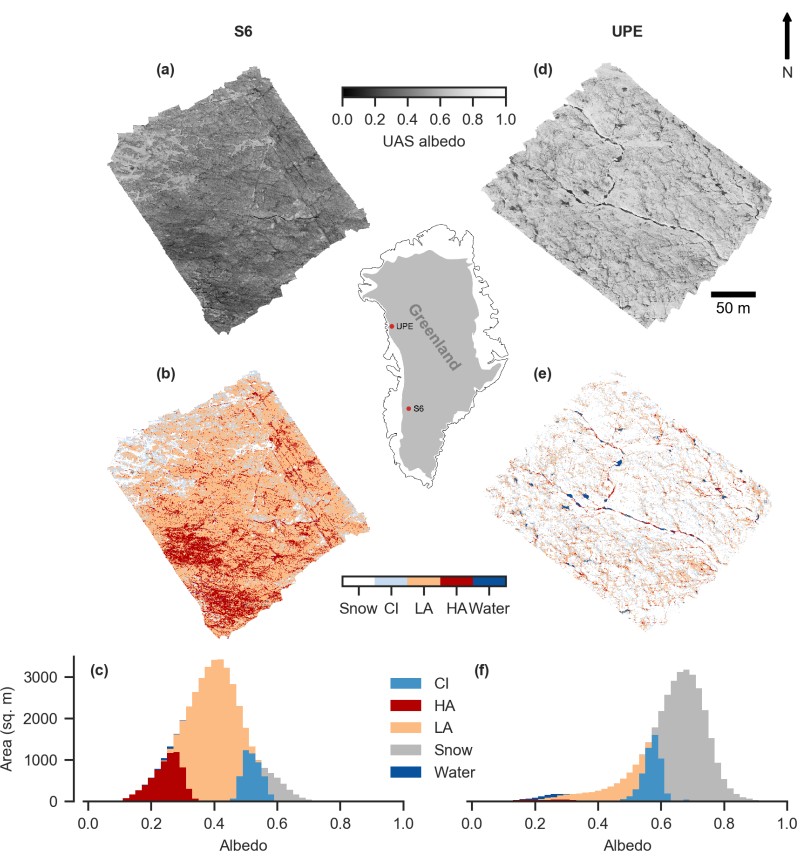

**Figure 1.** S6 (21 July 2017) and UPE (24 July 2018) UAS study area albedo and surface type. (a) UAS-measured albedo at S6, (b) UAS-measured albedo at UPE, (c) surface type classification at S6, (d) surface type classification at UPE, (e) stacked-bar histogram of surface type coverage at S6, (f) stacked-bar histogram of surface type coverage at UPE. CI: clean ice, LA: light algae, HA: heavy algae.

## 4.2 Topographic and hydrologic controls

The two sites were distinct in their local topography and hydrology. S6 had an average slope of 5º. Topographic features within the area principally consisted of (1) a ∼0.3 m wide ice-incised supraglacial stream and (2) a few isolated small (<2 m$^2$) ice rises up to ∼0.2 m high. After detrending (Sect. 3.1) 99% of the area had topographic variability of <±0.05 m and 54% of the area was within ± 0.01 m. The ice surface to ∼300 m up-slope of the area was flatter and had several small moulins, reducing the area contributing to local flow. The shallow and ephemeral arterial hydrological pathways present across the study area during July were likely the result of a constant slope and negligible meltwater routed from up-slope, reducing frictional stream incision (Ferguson, 1973). However, during June, winter snowpack retreat caused significant ephemeral sheet flow of water



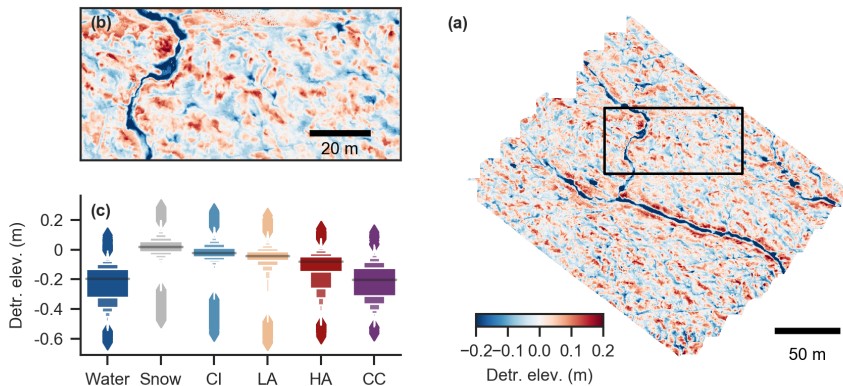

**Figure 2.** Surface topography variability at UPE. (a) Detrended elevation ('roughness'). Black box delineates the area shown in panel b. (b) Zoomed detail of detrended elevation, showing incised supraglacial stream and the pattern of local topographic highs and lows. (c) Letter-plots of detrended elevation for each surface type, illustrating median (black line), distribution of elevation values (boxes) and outliers (diamonds) within each category (Hofmann et al., 2011), computed from whole area shown in (a).

through the study area and caused algal cell re-distribution until up-slope crevasses and moulins opened to route meltwater away englacially. This was likely important in distributing concentrated algal blooms growing in local niches over a wider area given that glacier algae lack a flagellated life stage and so are not independently motile (Williamson et al., 2019).

Observations at UPE where there was substantial local surface roughness (Fig. 2a,b) showed that the higher the biomass

5 loading (from CI, through LA to HA), the lower the local elevation of the associated surface was (Fig. 2c). There are at least two possible reasons for the concentration of heavy algae in local depressions. One is entrainment and transport of algal cells in topographically higher areas by meltwater; once the competence of the meltwater flow drops in local depressions then the impurities will be deposited. Another is that local depressions favour near- or at-surface availability of meltwater through ponding, especially if a weathering crust is well-developed at topographic highs. Surface meltwater reduces albedo

10 (Zuo and Oerlemans, 1996; Greuell, 2000; Greuell et al., 2002) which results in favourable growth conditions for glacier algae (Williamson et al., 2018), further reducing albedo and amplifying surface ablation.

There are strong indications that the local topography and near-surface hydrology at UPE resulted in a different surface state to S6. The shallower slope (1º) than at S6 is likely to favour the evolution of perched meltwater ponds (Fig. 1e) as the lower gravitational potential is less conducive to runoff. Meanwhile, meltwater generated further upglacier flows through the

15 area in streams incised to ~0.6 m below the mean surface elevation (e.g. the stream running from north-west to south-east through study area, Fig. 2a). Arterial meltwater pathways are thus likely to persist inter-annually as little melting had occurred in the 2018 melt season prior to our measurements (Sect. 4.1). This stream-dominated hydrological regime likely reduces the movement of microbial cells suspended in meltwater through the weathering crust (Irvine-Fynn et al., 2012; Cook et al., 2016; Christner et al., 2018) compared to S6. A stream-dominated regime therefore also favours complex spatial and temporal



patterns of albedo where most ice is weathered, persistently bright and strongly scattering due to minimal sub-surface melt water, punctuated by low-albedo melt ponds and concentration of LAIs and water in topographic lows.

### 4.3 Change in physical surface properties

When a weathering crust develops then opportunities for volume scattering are increased, raising albedo. Conversely, weather-
ing crust removal decreases scattering opportunities, lowering albedo. Weathering crust status can be diagnosed through repeat measurements of reflectance in the NIR part of the spectrum made by our UAS, centred on 840 nm. Absorption by LAIs such as glacier algae is concentrated in the visible part of the solar spectrum while at 840 nm the albedo-reducing effect of glacier algae is negligible (Cook et al., 2017, 2019b, a; Williamson et al., 2019), and so variations in the NIR are primarily due to changes in near-surface ice properties. Whilst there may be some residual albedo reduction attributable to black carbon
(Warren, 1984) we believe that the dominant signal retrieved at 840 nm by our UAS is indicative of the weathering crust state, inclusive of ice grain sizes, ice density, porosity and interstitial and surface meltwater.

Our time series of UAS images from S6 allows us to investigate this pheonomenon in more detail. There was a widespread increase in 840 nm reflectance between 20 July and 21 July (Fig. 3a). True-colour composites indicate a transition from wet, polished and impermeable ice surfaces (Fig. 3c,f) to drained, whiter ice with meltwater draining through the porous near-surface
(Fig. 3d,g). This change was coincident with the surface energy balance returning to a shortwave-dominant regime following 4 days of dramatically reduced net shortwave radiation (Fig. 5a) and rainfall. Radiative fluxes dominated the energy budget between 21 and 22 July (Fig. 5a), and there were no further systematic changes in NIR reflectance (Fig. 3b) or true-colour composites (Fig. 3d,e). These findings are consistent with previous studies showing that weathering crust development versus decay is controlled primarily by the relative dominance of radiative or turbulent fluxes in the surface energy budget (Muller and
Keeler, 1969). Further, the reduction of albedo by rainfall through weathering crust stripping means that the melt-generating potential of cyclonic moisture intrusions which have been shown to account for $\sim 40\,\%$ of total precipitation over Greenland (Oltmanns et al., 2019) is likely to be higher if this rainfall-albedo feedback is accounted for in regional climate models.

### 4.4 Surface classification change through time

Repeat UAS acquisitions at S6 showed that the proportional coverage of different surface classes varied significantly from one
day to the next (Fig. 4). The reduction in snow and CI between 15 July and 20 July was caused by rainfall and high winds on 18 and 19 July which resulted in high sensible heat fluxes (Fig.5a) and rapid surface melting on 19 July despite low net shortwave radiation (Fig 5b). Rainfall caused widespread reduction of the thickness of the porous near-surface weathering crust layer and transient cryoconite hole melt-out, dispersing cryoconite granules and darkening the surface further (Shimada et al., 2016; Takeuchi et al., 2018). On 20 July only 5% of the surface was CI, compared to ∼10% on 15 July, with the majority of the area
(87%) classified as LA or HA. However, the data used to train our classifier has few examples of CI which are dark due to very thin or absent weathering crusts and so it is likely that some CI surfaces may have been mis-classified as LA. Subsequently, CI coverage increased on 21 July and was associated with a 9% increase in albedo (Fig. 5c) and regrowth of the weathering crust (Fig. 3).

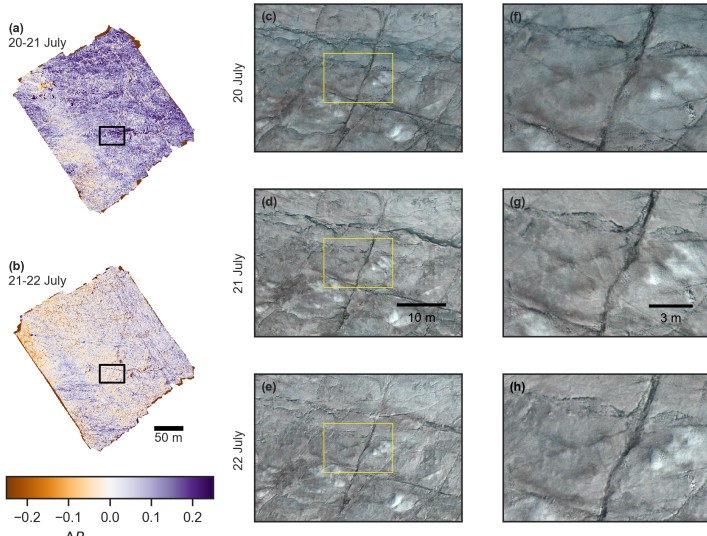

**Figure 3.** Weathering crust evolution. (a) Change in 840 nm reflectance between 20 and 21 July 2017: positive values indicate an increase in reflectance from 20 to 21 July. (b) as (a) but for 21–22 July change. (c-e) RGB-true-colour composites of surface within black rectangle shown in panels (a) and (b). (f-h) Zoomed RGB-true-colour composites of surface within yellow rectangle in panels c-e. (c,f) 20 July 2017, (d,g) 21 July 2017, (e,h) 22 July 2017.

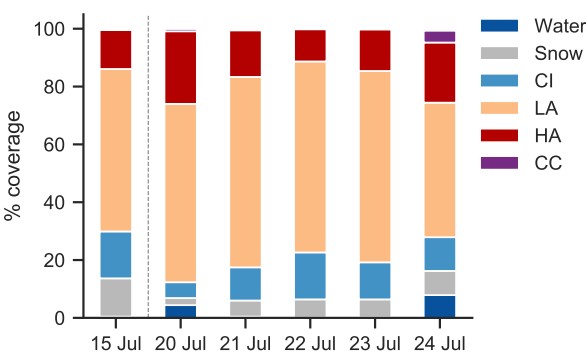

**Figure 4.** Percentage coverage of each surface type through time at S6.

From 21 to 23 July there was relatively little change in proportional surface cover. However, from 23 to 24 July there was a substantial increase in HA, together with the appearance of water and cryoconite and a 10% albedo reduction (Fig. 5c).

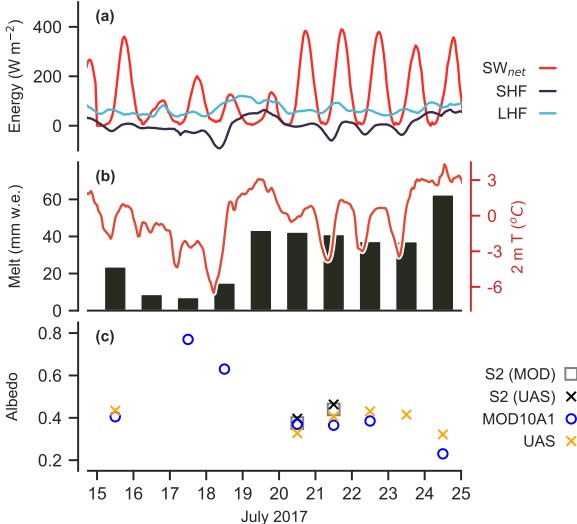

**Figure 5.** Time series of energy fluxes, surface melt rates and sensor albedos. (a) Energy balance components derived from surface energy balance model (Sect. 3.6). (b) Daily melt rate in mm water equivalent (bars) estimated with surface energy balance model and 2 m air temperature (line) from IMAU S6 AWS. (c) Albedo measured by UAS, S-2 within UAS area, S-2 within the MOD10A1 pixels, and mean MOD10A1 albedo.

Furthermore, variable illumination conditions during the 24 July flight over the western half of the study area caused overestimation of reflectance, likely favouring classification as CI, and so we probably did not capture the full magnitude of surface darkening.

The apparent increase in HA coverage on 24 July was probably not driven entirely by algal growth. Population doubling
5   times are estimated to be 5 d (Williamson et al., 2018), longer than the 1 d here. Indeed, LA coverage declined on 24 July while CI remained constant, whereas we would expect both LA and HA to increase in the case of widespread population growth. Instead, cells in LA areas may have been mobilised by the abundant surface meltwater and then deposited downslope in higher concentrations: air temperatures stayed above $0^{\circ}$C overnight from 23 to 24 July (Fig. 5b) associated with higher sensible heat fluxes (Fig. 5a), causing the most daily melting of the observation period (Fig. 5b). Furthermore, the sensible heat flux increased
10   the proportion of surface melting relative to sub-surface melting by shortwave penetration, likely thinning the weathering crust and further increasing the amount of liquid meltwater available on the surface, reducing albedo and increasing the likelihood of misclassification as HA. However, we note that our classification approach relies on coarse surface categories. Any LA ice patch loaded with algae towards the upper bounds of $10^3$ cells only needs a relatively small amount of growth to become loaded with $10^4$ cells found in HA samples (Sect. 3.4), and so in some pixels the algal population need not have doubled in order to
15   switch from LA to HA. We therefore cannot rule out the role of algae in causing daily surface type changes.

These findings illustrate that there are two principal reasons why surface classes might change through time: (1) algal growth (and removal, for instance by flushing by meltwater), and (2) physical changes which result in (mis-)classification. We cannot





uniquely distinguish between changes caused by algae versus by the weathering crust. First, algal growth is associated with enhanced melting, which reduces the thickness of the weathering crust and liberates liquid water and nutrients, stimulating further growth (Cook et al., 2019a). Second, changes in weathering crust optics occur beneath the algae, so any diagnostic algal feature present in our UAS images may change as the surface microtopography consitituting the cell habitat changes.

Third, there is uncertainty in spectral biomarkers unique to glacier algae. Theoretically, a simple band ratio, spectral feature identification or spectral mixing technique could be used to detect glacier algae as has been achieved for snow algae (Takeuchi et al., 2015; Painter et al., 2001; Huovinen et al., 2018). However, absorption by *Mesotaenium berggrenii* and *Ancylonema nordenskiöldii* (Williamson et al., 2019), the species found on the GrIS, is dominated by phenolic compounds that absorb strongly across the visible wavelengths (Williamson et al., 2018; Remias et al., 2012) and obscure potentially diagnostic spectral fea-

tures associated with other algal pigments (Cook et al., 2017, 2019b). A subtle absorption feature related to Chlorophyll-a is sometimes detectable using high spectral resolution measurements but is not visible in our multispectral imagery.

### 4.5   Upscaling to satellite scales

Our measurements at S6 were undertaken coincident with clear-sky observations by S-2 and MODIS MOD10A1. There was generally close agreement between UAS and satellite-derived albedo measured at S6 (Fig. 5c). We attribute discrepancies to

unavoidable differences between the radiometric calibration and narrowband-broadband conversion techniques and the different degrees of spatial integration. Nevertheless, the direction and magnitude of albedo change between the UAS and S-2 showed good agreement, whilst in general the UAS and MOD10A1 agreed on the direction of albedo changes (Fig. 5c). In the following section we use our UAS data to understand variability in surface type and albedo measured by S-2 and MOD10A1.

### 4.5.1   Characterisation of sub-S-2-scales

Sensor spatial resolution is important for algae detection. Classified S-2 data (Fig. 6a,b) shows that only CI and LA were identified at 20 m resolution, whereas at 5 cm resolution UAS imagery clearly showed frequent patches of HA within any arbitrary 20 by 20 m sub-area (Fig. 1b).

15% of S-2 pixels covering the UAS area changed from LA to CI between 20 July and 21 July. We used our UAS data to examine changes in surface class within each S-2 pixel (Table 1). The differences between those S-2 pixels which changed

class versus those which did not were small and S-2 pixels which transitioned to CI continued to be algae-dominated. This demonstrates that the patch dynamics of algal blooms, spatio-temporal variations in snow melt, weathering crust dynamics and surface roughness at sub-S-2-pixel scales ($\sim$1–10 m) are highly relevant for the interpretation of S-2 measurements and hence the attribution of surface melting to specific processes.

Spatial aggregation favours measurement of the mean surface properties. Our measurements suggest that under predomi-

nantly snow-free conditions then for an S-2 pixel to be classified as LA, the majority (>80%) of of the pixel needs to be covered in algae, with a significant amount of HA to compensate for the impact of residual CI areas upon the spatial average. We expect that 100% coverage by LA would also be sufficient to identify algal coverage at S-2 scales but we cannot show this with our data.





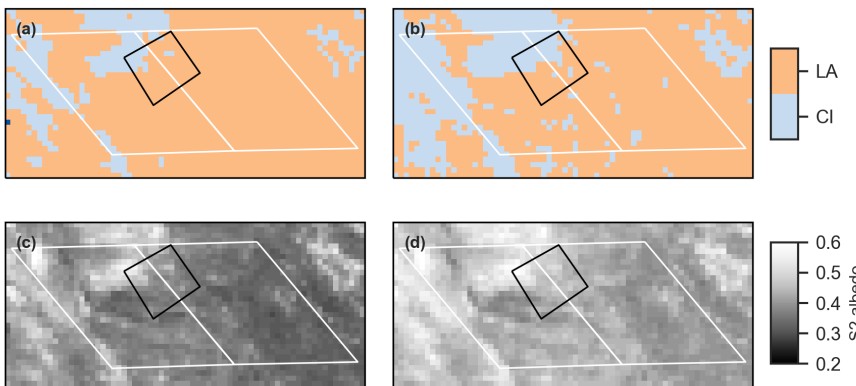

**Figure 6.** Observations from S-2 in UAS and MODIS areas. (a,b) Surface type classification from S-2. (c,d) Albedo from S-2. (a,c) 20 July, (b,d) 21 July. White rectangles indicate 500 m MODIS sinusoidal grid pixels covering study area; black rectangle indicates UAS study area.

| | LA ⇒ LA | | | LA ⇒ CI | | |
|---|---|---|---|---|---|---|
| | 20 July | | 21 July | 20 July | | 21 July |
| CI | 1 % | ⇑ | 10 % | 5 % | ⇑ | 18 % |
| LA | 59 % | ⇑ | 65 % | 73 % | ⇓ | 68 % |
| HA | 33 % | ⇓ | 18 % | 15 % | ⇓ | 7 % |

**Table 1.** Changes in the sub-S-2-pixel proportional coverage of the main surface classes from 20 to 21 July, aggregated for those S-2 pixels which did not change class (LA⇒LA) compared to those which did (LA⇒CI). Vertical arrows show direction of change between days.

Under reduced shortwave conditions on 20 July there was some evidence of a bi-modal albedo distribution within CI S-2 pixels (Fig. 7a). Once shortwave-dominant conditions returned the albedo distribution became more gaussian (Fig. 7b). In contrast, the albedo distribution within LA S-2 pixels exhibited unimodal gaussian characteristics on both days (Fig. 7c,d). Nevertheless, within the LA class there was an appreciable shift from 20 to 21 July to a larger range in sub-S-2-pixel albedo 5 (Fig. 7c,d), highlighting significant variability in sub-pixel albedo. Between 20 and 21 July, 91% of the UAS study area remained the same or increased in albedo (Fig. 7e). Areas in which albedo declined already had low albedo (as expressed by the colour of each curve in Fig. 7e), while the surfaces which increased in albedo already had high albedo.



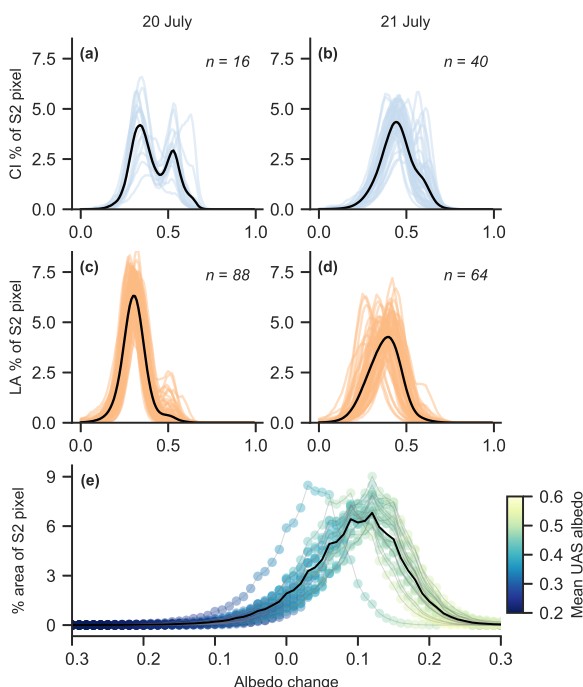

**Figure 7.** Sub-S-2-pixel albedo distributions derived from UAS measurements. (a-d) Distributions (one line per S-2 pixel), with black line indicating mean albedo distribution; albedo on x-axis. (a) Clean-ice pixels on 20 July, (b) Clean-ice pixels on 21 July, (c), Light-algae pixels on 20 July, (d) Light-algae pixels on 21 July. (e) Distributions of albedo change in the pixels which changed class between 20 July and 21 July (one line per pixel), in 0.02 bins. Colour of each bin corresponds to mean albedo of pixels in the bin on 21 July.

It is clear that S-2 estimates of algal growth presence are conservative. This is consistent with Cook et al. (2019b) who found much higher HA coverage in UAV imagery than S-2 imagery due to spatial integration which captures the mean reflectance of the whole area of interest. This suggests that their estimates of spatial coverage by algae over the GrIS western ablation zone and their derived estimate of total runoff attributed to ice algal growth (6–9 %) are likely to be conservative. Furthermore,

5 like in our UAS imagery, detection of algae by S-2 is likely to be confounded by changes in the weathering crust which cause optical changes of similar or greater magnitude than those attributable to glacier algae alone.

### 4.5.2 Characterisation of sub-MODIS pixel scales

The daily MODIS albedo product, MOD10A1, has a coarse spatial resolution of 500 m and is known to disagree with smaller-scale in-situ measurements of albedo at automatic weather stations, especially in the ablation zone (Ryan et al., 2017), which

10 may have ramifications for melt rate calculations that depend on MOD10A1 albedo. We used S-2 observations to examine sub-MODIS-pixel MOD10A1 albedo distributions in the same way that we used UAS data to examine sub-pixel S-2 albedo



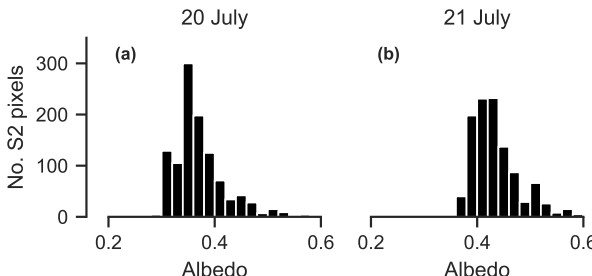

**Figure 8.** Histograms of S-2-derived albedo within the two MODIS pixels covering the UAS survey area on (a) 20 July and (b) 21 July.

distributions. For both days of S-2 observations, we examined all $20\times20$ m S-2 pixels that fell inside two MODIS pixels at S6 (Fig. 6).

S-2 albedo within the two MODIS pixels was non-normal and left-skewed on both days of S-2 overpass (Fig. 8). Despite substantial sub-MODIS-pixel changes in albedo there was no net change observed in the mean MOD10A1 albedo of the two

pixels (Fig. 3a). Examination of each MODIS pixel separately (Fig. 6) showed that 17% of the western pixel changed from LA to CI yet, in contrast, MOD10A1 indicated a 1% albedo decrease, while in the eastern pixel 7% of the area changed from LA to CI yet no albedo change was detected by MOD10A1. Albedo increases were measured by S-2 in both MOD10A1 pixels. This demonstrates that low spectral and spatial resolution MODIS imagery fails to resolve spatio-temporal patterns of albedo at the surface and so it cannot be used to attribute melting to specific processes such as weathering crust dynamics, biological

growth and decline, impurity accumulation and supraglacial hydrology.

To estimate the impact of non-normal sub-MODIS-pixel albedo distributions on melt rates we ran our energy balance model in 0.01 albedo increments, with fluxes fixed to those observed at S6 on 21 July at 13:00 local time, to derive an hourly melt rate for each albedo value in the distribution. We then applied these melt rates to each S-2 pixel within the two MODIS pixels to estimate the melt flux between 13:00 and 14:00. On 20 July, the distribution-derived melting caused by net shortwave radiation

was 241 $m^3$, whereas using the mean albedo computed from all S-2 pixels it was 236 $m^3$ w.e. On 21 July melting was estimated as 217 $m^3$ w.e. and 213 $m^3$ w.e respectively. The sub-MODIS-scale skew in albedo distribution therefore has a small but non-negligible ($\sim$2%) difference on estimated surface melting and warrants further investigation over wider spatial and temporal scales.

## 5   Conclusions

Glacier algae are ubiquitous in the western GrIS ablation zone. Their local distribution across the ice surface is principally a function of local topography and the characteristics of the surface hydrological network. Rougher surfaces yield local depressions in which concentrations of algae tend to be higher, suggesting that environmental conditions for growth — especially liquid meltwater presence — are met more readily in these areas and/or that cells which have grown elsewhere can be mo-



bilised and then deposited further downstream. These bio-physical characteristics result in significant albedo variability when compared to smoother ice surfaces where glacier algae tend to be distributed more homogeneously.

The distribution and concentration of algal blooms at local scales changes significantly from one day to the next. However, algal population sizes require several days to double and therefore apparent increases in high algal coverage from one day to the

next are more likely to principally be the result of local mobilisation and re-deposition in concentrated patches by supraglacial meltwater flow. Furthermore, whilst glacier algae are potent albedo reducers, daily albedo changes are predominantly associated with physical weathering crust changes controlled by the surface energy budget. The optics of the weathering crust are so dominant over other albedo-affecting processes that under high turbulent heat fluxes the albedo is principally determined by the state of the weathering crust. Only under shortwave-dominant energy conditions can a weathering crust develop, enabling

LAIs to exert more control upon albedo both directly and by modifying the optics of the underlying ice surface via enhanced melting at patch scales.

Upscaling of our observations to satellite sensor scales shows that Sentinel-2 is conservative in its detection of glacier algae and so retrievals of algal biomass by Sentinel-2 are likely to be under-estimated, especially under meteorological conditions that enable widespread development of a weathering crust. Under shortwave-dominant energy conditions, albedo over 20 m

scales (sub-S2-pixel) is generally uni-modal and unskewed and so is representative of sub-pixel albedo variability. At 500 m scales, MOD10A1 does not always capture widespread albedo changes measured by other sensors. Sub-MOD10A1 albedo distributions were left-skewed over our bare-ice study area, which is equivalent to a ∼2% under-estimate in melting derived from surface energy budget calculations which use only MOD10A1 albedo. Future research should seek to further constrain weathering crust processes and their controls upon albedo, and should favour use of higher spatial resolution albedo data in

heterogeneous ablation zones.

*Code and data availability.* During the Discussion phase, the code underlying the processing and analysis can be found at https://github. com/atedstone/GrIS_ice_albedo_variability.git. The trained classifiers, processed UAS data, ground spectroscopy data, algal cell counts and classified Sentinel-2 data can be found at https://www.dropbox.com/sh/yjpwi5kdhyg2vt8/AAAOz4UwIYJ-UOgSKG_HAUkMa. Digital Object Identifiers will be created for the code and datasets upon acceptance of the final manuscript. Unprocessed UAS data are lodged with the

UK Polar Data Centre (S6: shortdoi:10/c72x, UPE: shortdoi:10/c72z). UPE_U AWS data were provided by the Programme for Monitoring of the Greenland Ice Sheet (PROMICE) and the Greenland Analogue Project (GAP) through the Geological Survey of Denmark and Greenland (GEUS) (http://www.promice.dk) and S6 AWS data were provided by the Institute for Marine and Atmospheric Research, Utrecht (IMAU, https://www.projects.science.uu.nl/iceclimate/aws/). MODIS MOD10A1 data were provided by the National Snow and Ice Data Center (https://nsidc.org/data/mod10a1) and Sentinel-2 data were provided through Sinergise (https://www.sinergise.com) by the European Space

Agency SENTINEL Program (http://sentinel.esa.int).



# Appendix A: Overpass times

**Table A1.** Times of data acquisition by UAS, S-2 and MODIS (local time, UTC-2). Asterisk indicates variable illumination conditions during UAS flight.

| Date | UAS | S-2 | MODIS |
|---|---|---|---|
| 15 Jul | 11:00 | - | 13:40 |
| 20 Jul | 12:30 | 12:59 | 12:20 |
| 21 Jul | 15:10 | 13:19 | 13:05 |
| 22 Jul | 10:00 | - | 13:45 |
| 23 Jul | 11:00 | - | 12:50 |
| 24 Jul | 13:00* | - | 13:35 |





## Appendix B: Classifier confusion matrices

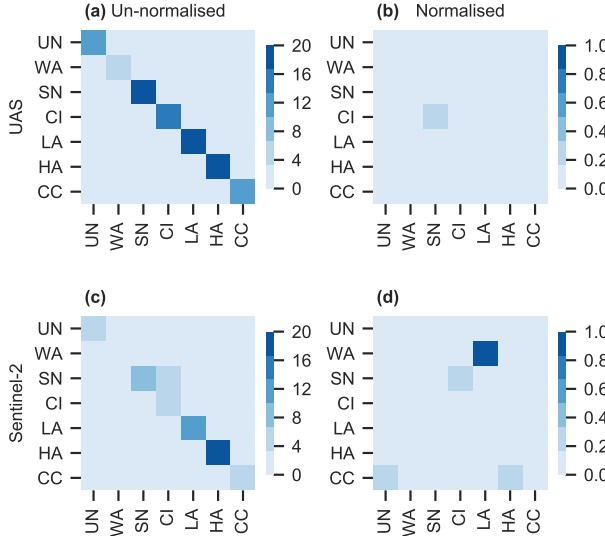

**Figure B1.** Confusion matrices and normalised confusion matrices for the Random Forests models applied to the UAS (a,b) and Sentinel-2 (c,d) data. Confusion matrices show predicted class on y-axis and actual class on x-axis. The scores at the intersections show the frequency of instances.





## Appendix C: Algal cell counts at UPE

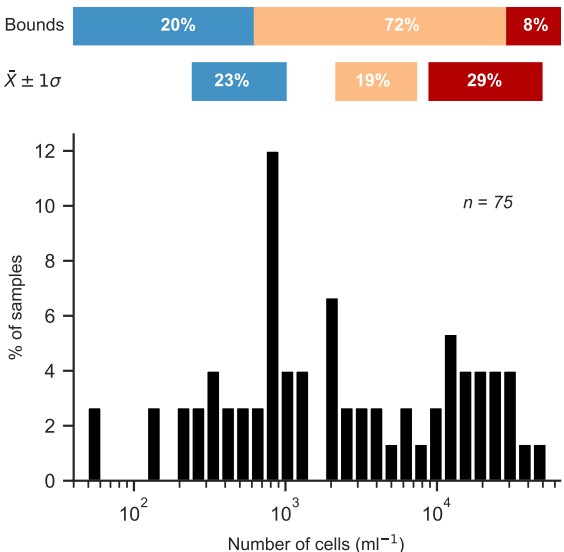

**Figure C1.** Histogram of cell counts undertaken at UPE on 26 July 2018. The horizontal bars illustrate the range of CI (blue), LA (orange) and HA (red) by two different metrics: (a) 'bounds', using the boundaries of CI < 625 $\mathrm{cells\,ml^{-1}}$, HA > $2.9{\times}10^4$ $\mathrm{cells\,ml^{-1}}$, with LA corresponding to the values between these boundaries, and (b) $\bar{X} \pm 1\sigma$, which corresponds to the abundance ranges of the surface type classes from S6 reported by Cook et al. (2019b) which were used to train the surface classifier used in this study (Sect. 3.4). Percentage values refer to the number of surface samples which fall into each of these categories.

Seventy-five biological samples taken at randomised coordinates within the UPE survey area (Sect. 4.1) revealed the widespread presence of glacier algae (Fig. C1). Whether using the cell abundance ranges defined with S6 measurements Cook et al. (2019b) or using the mean S6 cell abundances to define boundaries between different surface types, it is clear that cell abundances representative of LA and HA coverage were present on the bare ice surface. Under the bounds-based approach, which enables us to include all of our samples in estimating proportional surface type cover, 80% of the UPE survey area was algae-covered.





*Author contributions.* AT and JC designed the study. JC built and tested the UAS. AT, JC, SH, CW, JM and TG collected field data. AT post-processed the UAS imagery. JC and AT developed the surface type classification approach. CW counted the algal cells sampled at the UPE site. AT analysed the data, produced the figures and wrote the manuscript. JC and CW wrote sections of the manuscript. All authors commented on the findings and edited the manuscript.

5 *Competing interests.* The authors declare that there are no competing interests.

*Acknowledgements.* We acknowledge funding from the UK National Environment Research Council Large Grant NE/M021025/1 'Black and Bloom'. JC acknowledges the Rolex Awards for Enterprise, National Geographic and Microsoft ('AI for Earth'). SH acknowledges the European Research Council grant agreement no. 694188 'GlobalMass'. TG acknowledges the Gino Watkins Memorial Fund and Nottingham Education Trust. TI acknowledges NERC NE/M020991/1 and Leverhulme Trust Fellowship (RF-2018-584/4). In addition to the authors, the
10 'Black and Bloom' project team comprises A. Anesio, J. Bamber, L. Benning, E. Hanna, A. Hodson, A. Holland, S. Lutz, J. McQuaid, M. Nicholes, E. Sypianska and M. Yallop.



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
