# Peer review of "Algal growth and weathering crust state drive variability in western Greenland Ice Sheet ice albedo"

_The Cryosphere, 2019_

## Referee Comment (RC1) · Anonymous Referee #1 · 16 Aug 2019

Tedstone et al. investigate the importance algal growth and weathering crust formation on bare ice albedo variability in western Greenland. They use observations from a field camp in 2017 to describe the bare ice surface and optical satellite imagery provided by MODIS and Sentinel-2 to demonstrate that coarser resolution satellite imagery will underestimate algal presence. They also find that bare ice surfaces have a left-skew albedo distribution at the scale of MODIS pixel which suggests that when MODIS data are used for energy balance modelling, meltwater production may be underestimated by 2%. The combination of field observations with satellite imagery to investigate the scale-gap between point and pixel albedo measurements enables new insights into Greenland's bare ice surface that should be considered when modelling the surface

mass balance of the ice sheet. The manuscript therefore fits within the scope of The Cryosphere and deserves to be published.

My one major comment on the manuscript is that, considering the availability of high-resolution DEMs for both sites, the relationship between albedo and topography is not investigated in great detail. For example there is only one figure showing surface topography and that only shows the UPE site. At least consider adding more panels to Fig. 2 showing the surface topography at S6. Better would be include some additional analysis which demonstrates that low albedo pixels are more likely to be found in topographic depressions. This would provide some evidence to back-up the qualitative statements in the conclusions (P8 L4-11, P15 L20- to P16 L1-2).

Below are some more specific suggestions that the authors may find useful to consider.

P2 L1: Consider adding "into the ocean" after "directly" to clarify for non-specialists.

P2 L9: The Box et al. (2012) paper does not appear to have mapped bare ice so cannot have attributed the importance of snowpack melting, consider removing this reference.

P16 L3: Consider quantifying this statement with a percentage change.

Figure 2 Slightly confusing that panel a is on the right. Consider switching to the left of b and c.

Figure 3 Consider adding some lines from the the top left and bottom left of the yellow boxes to the top left and bottom left of panels f, g and h to make it clearer that these are zoomed versions of the same image.

Figure 6 Consider adding dates to panels a and b so it's more obvious that these are the same area on two different days.

———————————————

---

## Referee Comment (RC2) · Anonymous Referee #2 · 19 Aug 2019

This paper aims at investigating Greenland Ice Sheet surface albedo variation from algal growth and weathering crust using UAS, Sentinel-2, and MODIS albedo datasets. The paper is well-written but there are two major issues that need to be addressed before it can be considered publication in The Cryosphere. 1. There is some confusion about the use of reflectance and albedo, which may cause potentially large errors in the "albedo" comparison. For example, both the UAS camera and the Sentinel-2 observations are multispectral bi-directional reflectance, not same as the albedo, even after the normalization with the Spectralon, atmospheric correction, and narrow-to-broadband conversion. To calculate albedo from reflectance, Bi-directional Reflectance Function (BRDF) needs to be taken into account. The MODIS snow albedo product (MOD10A1)

[Figure]

is derived using surface reflectance and pre-assumed BRDF shapes. Recent studies have shown that snow/ice BRDF effects cannot be ignored especially under high solar zenith conditions (Gatebe and King, 2016; Jiao, et al., 2019). UAS reflectances from the backward and forward viewing directions can have a large difference (resulting from the BRDF effects), leading to the bi-modal instead of the unimodal albedo distribution.

Gatebe, C. and King, M. (2016). Airborne spectral BRDF of various surface types (ocean, vegetation, snow, desert, wetlands, cloud decks, smoke layers) for remote sensing. Remote Sensing of Environment, 179, 131-148

Jiao, Z. et al. (2019). Development of a snow kernel to better model the anisotropic reflectance of pure snow in a kernel-driven BRDF model framework. Remote Sensing of Environment, 221, 198-209

2. One of the key findings is that albedo datasets at different spatial resolutions can have different results; this could also be a result of artifacts from data selection/preprocessing. For example, surface albedo increases as solar zenith becomes larger, so for the same object surface albedo will be the lowest at local noon. The UAS data obtained at Jul 20th had the lowest solar zenith and thus the lowest surface albedo, so the albedo obtained at Jul 21st and 22nd would be larger than albedo at Jul 20th. Since the solar zenith angles for the images obtained at Jul 21st and 22nd would be similar, so would be their albedo values. In comparison, the Sentinel-2 and MODIS would have much smaller difference in albedo values on Jul 20th and 21st. Solar zenith corrections are needed here before any further analysis on albedo changes can be carried out.

---

## Referee Comment (RC3) · Anonymous Referee #3 · 17 Sep 2019

Tedstone et al. presented a study using field data and remote sensing data to analyze how algae and weathering crust change Greenland Ice Sheet albedo over two ablation sites in west Greenland (namely S6 and UPE). It is concluded in the abstract that ice albedo is affected by both light-absorbing impurities (not only algae??? please clarify) and physical ice processes (specifically weathering crust??? please clarify), and there is a spatial scale dependency in albedo measurements which should be considered. I found this manuscript very similar to a very recent TC discussion paper by Cook et al. (2019) 'Glacier Algae accelerate melt rates on the south-western Greenland Ice Sheet'. I think the authors need to clarify the difference of this manuscript from that paper, given the same datasets, methods, and surface classification results at S6 site.

[Figure]

The title needs to be improved. It is not quite right to say that algal growth and weathering crust drive the albedo variability of the Greenland Ice Sheet given the importance of snow metamorphism on the albedo of the accumulation zone. Since the authors only discussed two sites at the ablation zone in west Greenland. Please specify west Greenland and bare ice albedo.

Sections 3.1, 3.2, 3.3, and 3.4 are almost as same as the Cook et al paper. Although the Cook et al paper is cited, it is inappropriate to repeat the same content from another independent paper unless the authors clarified the relationship and difference between these two papers.

The surface type classification section is a critical part for analyzing the changes of algae and weather crust (I guess the authors are trying to focus on these two factors). However, why don't the authors include weathering crust as a surface type when using the random forest method to classify the UAS image and Sentinel-2 image. Again, the authors directly used the surface type classification results from Cook et al. (2019) which didn't consider weathering crust. Besides, it is not appropriate to make statements based on the results from another under-review paper (Cook et al. 2019). What's the criteria to separate the high algae surface from low algae surface? Using thresholds? How to define the threshold? The high vs low algae sound very arbitrary.

As the authors mentioned that the remote sensing data they were using don't contain spectral signature of algae, in this case, how could the algae surfaces be classified? In other words, how to separate them from other impurities?

Page 11 line 4-15, this part is very unclear, rephrasing is necessary.

Although the authors emphasized the importance of weathering crust on surface albedo, but I didn't find detailed quantitative analysis about this subject? Section 4.3 reads quite speculative. Any references to use 840nm to identify the weathering crust?
The authors aimed to analyze the impact of algae and weathering crust on Greenland ice sheet albedo, but the datasets are limited to two specific sites. Discussion about the generalization of those two specific sites to larger spatial scale is necessary.

Regarding the scale problem, particularly the impact of scale on melt flux estimation (page 15 line 11-18), did the authors use the actual MODIS albedo to estimate the melt flux? Please clarify. It seems that all the melt flux estimates are based on Sentinel-2 albedo, one scenario is to use Sentinel-2 albedo for each individual Sentinel-2 pixel, another scenario is to calculate an average Sentinel-2 albedo over a MODIS pixel scale. I don't think this comparison is fair, what the difference between the Sentinel-2 averaged albedo and the real MODIS albedo? Without considering this, "the ∼2% underestimate in melting derived from surface energy budget calculations which use only MOD10A1 albedo" (in abstract and conclusion) is wrong. Besides, using only two MODIS pixel to make this statement is not sound. The authors should consider conducting the calculation over a large scale, since Sentinel-2 image and MODIS image can cover a large area instead of two MODIS pixels.

---

## Author Comment (AC1) · 27 Sep 2019

**Response to RC1, 16 August 2019**

*Tedstone et al. investigate the importance algal growth and weathering crust formation on bare ice albedo variability in western Greenland. They use observations from a field camp in 2017 to describe the bare ice surface and optical satellite imagery provided by MODIS and Sentinel-2 to demonstrate that coarser resolution satellite imagery will underestimate algal presence. They also find that bare ice surfaces have a left-skew albedo distribution at the scale of MODIS pixel which suggests that when MODIS data are used for energy balance modelling, meltwater production may be underestimated by 2%. The combination of field observations with satellite imagery to investigate the scale-gap between point and pixel albedo measurements enables new insights into Greenland's bare ice surface that should be considered when modelling the surface mass balance of the ice sheet. The manuscript therefore fits within the scope of The Cryosphere and deserves to be published.*

We thank R1 for their positive review and comments.

*My one major comment on the manuscript is that, considering the availability of high resolution DEMs for both sites, the relationship between albedo and topography is not investigated in great detail. For example there is only one figure showing surface topography and that only shows the UPE site. At least consider adding more panels to Fig. 2 showing the surface topography at S6.*

As we note in P7 L4-5, following detrending of the 5° slope, 99% of the area mapped at S6 had a topographic variability of <0.05 m, 54% of area within <0.01 m variability. This means that (a) there is no topographic information worth plotting like we did for UPE, and (b) no relationship between topography and albedo is evident because there is no local topographic variability.

*Better would be include some additional analysis which demonstrates that low albedo pixels are more likely to be found in topographic depressions. This would provide some evidence to back-up the qualitative statements in the conclusions (P8 L4-11, P15 L20- to P16 L1-2).*

To some extent, this is already present in the manuscript: Figure 2(c) shows the average detrended elevation of each surface type. It clearly indicates that progressively darker surface types are found in deeper local depressions. Nevertheless, we will undertake this additional analysis for the revised manuscript, most likely to be provided as a set of summary statistics rather than a figure.

*Below are some more specific suggestions that the authors may find useful to consider.*

*P2 L1: Consider adding "into the ocean" after "directly" to clarify for non-specialists.*

**We will implement in revised manuscript.**

*P2 L9: The Box et al. (2012) paper does not appear to have mapped bare ice so cannot have attributed the importance of snowpack melting, consider removing this reference.*

**Thanks, we will remove in revised manuscript.**

*P16 L3: Consider quantifying this statement with a percentage change.*

"The distribution and concentration of algal blooms at local scales changes significantly from one day to the next" – quote number and edit to m/s here.

*Figure 2 Slightly confusing that panel a is on the right. Consider switching to the left of b and c.*

Agree in principle. Originally the figure had panel a on the left, but it didn't look so neat – nevertheless we **will implement in revised manuscript.**

*Figure 3 Consider adding some lines from the the top left and bottom left of the yellow boxes to the top left and bottom left of panels f, g and h to make it clearer that these are zoomed versions of the same image.*

**Agree, we will try to apply proposed change to revised manuscript.**

*Figure 6 Consider adding dates to panels a and b so it's more obvious that these are the same area on two different days.*

**Agree, we will apply proposed change to revised manuscript.**

**Response to RC2, 19 August 2019**

*This paper aims at investigating Greenland Ice Sheet surface albedo variation from algal growth and weathering crust using UAS, Sentinel-2, and MODIS albedo datasets. The paper is well-written but there are two major issues that need to be addressed before it can be considered publication in The Cryosphere.*

We thank R2 for their review and their statement that the paper is well-written.

*1. There is some confusion about the use of reflectance and albedo, which may cause potentially large errors in the "albedo" comparison. For example, both the UAS camera and the Sentinel-2 observations are multispectral bi-directional reflectance, not same as the albedo, even after the normalization with the Spectralon, atmospheric correction, and narrow-to-broadband conversion. To calculate albedo from reflectance, Bi-directional Reflectance Function (BRDF) needs to be taken into account. The MODIS snow albedo product (MOD10A1) is derived using surface reflectance and pre-assumed BRDF shapes. Recent studies have shown that snow/ice BRDF effects cannot be ignored especially under high solar zenith conditions (Gatebe and King, 2016; Jiao, et al., 2019). UAS reflectances from the backward and forward viewing directions can have a large difference (resulting from the BRDF effects), leading to the bi-modal instead of the unimodal albedo distribution.*

*Gatebe, C. and King, M. (2016). Airborne spectral BRDF of various surface types (ocean, vegetation, snow, desert, wetlands, cloud decks, smoke layers) for remote sensing. Remote Sensing of Environment, 179, 131-148*

*Jiao, Z. et al. (2019). Development of a snow kernel to better model the anisotropic reflectance of pure snow in a kernel-driven BRDF model framework. Remote Sensing of Environment, 221, 198-209*

We agree that anisotropic scattering is an issue that needs to be resolved for remote sensing over glacier ice. However, in the scope of this study we are limited by a lack of bidirectional reflectance distribution functions appropriate for correcting reflectance measurements over ablating glacier ice. To our knowledge, no BRDF datasets for ablating glacier ice exist. The two examples provided in the comment only contain BRDF data for snow. Snow and glacier ice are structurally, texturally and optically distinct and it is not justifiable to modify glacier ice reflectance using snow BRDFs. This is especially true in our case because we use data from a highly heterogeneous surface that includes surface with high light-absorbing impurity loading, surface water, cryoconite holes, dispersed cryoconite, variable surface topography, patches of rotten snow and areas of thick weathering crust, all of which vary dramatically in their optical characteristics. Although no data are currently available, the scattering anisotropy of these surfaces is likely to vary greatly and be very different to dry snow.

We therefore consider the application of any existing BRDF model to be unhelpful, perhaps even increasing the uncertainty in our measurements. If the reviewer can point us to suitable BRDF data for the various surfaces in our UAV images we will be delighted to incorporate it into our study. We also reviewed the MODIS MOD101A1 Collection 6 User Guide and confirmed that MOD101A1 uses BRDFs optimised for snow to correct reflectance to albedo over ablating ice. MODIS MOD101A1 albedo products are therefore also likely subject to significant error over these surfaces – a problem that requires new empirical data to resolve.

Nevertheless, we appreciate R2's concerns on this topic and therefore propose adding the following sentence to the manuscript:

**"We caveat that empirical bidirectional reflectance distribution functions are not available for the surface types included in our analysis. While these surfaces are non-Lambertian and scatter light preferentially in the forward direction, causing sensors at nadir to underestimate albedo, there are no datasets we know of that can accurately correct reflectance values gathered at nadir. We therefore omit a BRDF correction as existing BRDF datasets cannot be confidently applied to our sample surfaces."**

*2. One of the key findings is that albedo datasets at different spatial resolutions can have different results; this could also be a result of artifacts from data selection/preprocessing. For example, surface albedo increases as solar zenith becomes larger, so for the same object surface albedo will be the lowest at local noon. The UAS data obtained at Jul 20th had the lowest solar zenith and thus the lowest surface albedo, so the albedo obtained at Jul 21st and 22nd would be larger than albedo at Jul 20th. Since the solar zenith angles for the images obtained at Jul 21st and 22$^{nd}$ would be similar, so would be their albedo values. In comparison, the Sentinel-2 and MODIS would have much smaller difference in albedo values on Jul 20th and 21st. Solar zenith corrections are needed here before any further analysis on albedo changes can be carried out.*

Our UAV flights all occurred within 2 hours of the local solar-noon in order to minimise solar zenith angle errors. The Sentinel-2 and MODIS overpasses also fell within this window.

As we outline elsewhere in this response to reviews, an underlying issue is that the optics of glacier ice are unconstrained. It is therefore not possible to accurate compute solar zenith angle corrections. Nevertheless, we have assessed the potential impact following Warren (1982, Reviews of Geophysics), which is applicable for snow surfaces. We use the formulation employed by the regional climate model MAR (Fettweis et al., 2017, The Cryosphere) to compute an approximate albedo correction as a function of the cosine of the solar zenith angle:

```
Difference = (0.64 - csza) * 0.0625
```

At S6, using 21 July 2017 as an example, cos(SZA) was 0.83 at 10:00 local, 0.94 at 12:00 local, 0.85 at 14:00 local (as calculated using

https://www.esrl.noaa.gov/gmd/grad/solcalc/azel.html). These correspond to corrections of -0.012%, -0.019% and -0.013% respectively, therefore a range of 0.7%. Thus, the difference in albedo as a function of solar zenith angle during the time period over which we sampled is negligible and can be dis-regarded.

**Response to RC3, 17 September 2019**

*Tedstone et al. presented a study using field data and remote sensing data to analyze how algae and weathering crust change Greenland Ice Sheet albedo over two ablation sites in west Greenland (namely S6 and UPE). It is concluded in the abstract that ice albedo is affected by both light-absorbing impurities (not only algae??? please clarify) and physical ice processes (specifically weathering crust??? please clarify), and there is a spatial scale dependency in albedo measurements which should be considered.*

In the abstract we conclude that "among LAIs, pigmented glacier algae are ubiquitous and cause surface darkening both within and outside the south-west GrIS 'dark zone'" (P1 L6-7). We later note that "pigmented glacier algae are the dominant agent of darkening amongst LAIs" (P2 L17). Previous work (Cook et al. 2019) concludes, for S6, that glacier algae, not 'dust', is the dominant LAI. Nonetheless, in the revised manuscript **we will remove 'that among LAIs' from P1 L6**, to further clarify that our focus is on glacier algae.

**We will also change 'physical ice processes' on L15 to 'weathering crust processes'**, reflecting the focus of our manuscript on the weathering crust specifically (rather than, for instance, the structural glaciological properties of the emerging ice).

*I found this manuscript very similar to a very recent TC discussion paper by Cook et al. (2019) 'Glacier Algae accelerate melt rates on the south-western Greenland Ice Sheet'. I think the authors need to clarify the difference of this manuscript from that paper, given the same datasets, methods, and surface classification results at S6 site.*

We have been up-front about methodological cross-over between this study and Cook et al. (2019). Both papers make use of data gathered at S6 in 2017 and use classifiers trained on field-spectra applied to UAS multispectral data. We already highlight this in the manuscript, e.g. P3 L6-9. However, there are several substantial and important aspects that are unique to this paper that are not touched upon by Cook et al. (2019). We are surprised that the reviewer did not pick up on the contrasting research question, scope, field sites, data, processing methodology and findings of the two papers.

Cook et al. (2019) only processed and examined one UAS image. Here we provide time-series analysis of multiple UAS flights over the ice-sheet surface for the first time. Cook et al. established the role of algae in darkening the ice surface and accelerating melt rates. However, here we study the spatio-temporal variability of the ice albedo, examine the feedbacks to the physical configuration of the ice (i.e. weathering crust development) and quantify inter-sensor and cross-scale correspondence between ground measurements and remote sensing platforms. Furthermore, we present and analyse new data from the UPE field site that was not mentioned at all in Cook et al. 2019. Our study is therefore distinct from previous work.

To ameliorate R3's concerns about dataset usage we propose adding the following at the end of the sentence on P3, L21:

**UAS imagery acquired on 21 July 2017 have been presented previously (Cook et al., 2019) but this study is the first to analyse the full time series of UAS imagery that we acquired at S6 and to present the imagery acquired at UPE.**

*The title needs to be improved. It is not quite right to say that algal growth and weathering crust drive the albedo variability of the Greenland Ice Sheet given the importance of snow metamorphism on the albedo of the accumulation zone. Since the authors only discussed two sites at the ablation zone in west Greenland. Please specify west Greenland and bare ice albedo.*

The existing title already specifies 'ice albedo' and so the title is correctly constrained regarding snow versus ice. **We can add 'western' to the title.**

*Sections 3.1, 3.2, 3.3, and 3.4 are almost as same as the Cook et al paper. Although the Cook et al paper is cited, it is inappropriate to repeat the same content from another independent paper unless the authors clarified the relationship and difference between these two papers.*

Please also see response to previous comment on this matter.

Sections 3.1, 3.2 and 3.3 contain significant amounts of information on the approach undertaken for the UPE site that was not included in Cook et al. (2019) as well as providing crucial information underpinning the of the rest of the manuscript and so we are reticent to remove details from these sections: for example, the removal of ice motion from the orthomosaics, DEM production and biological sampling at UPE are all new in this manuscript.

For Sect 3.4, we apologise that referencing error crept in: P5 L17 should reference Cook et al. (2019), not Cook et al. (2017). **We will correct this in the revised manuscript.**

Should the editor/reviewer explicitly request it, we would be amenable to reducing the details provided Sect. 3.4 and simply referring the reader to Cook et al. (2019). However, we need to be careful to preserve readability: this is the section where we introduce the surface categories which are referred to repeatedly in the remainder of the manuscript.

Lastly, we note that the classifier used in this manuscript was trained especially for this manuscript and is not precisely the same as the one used in Cook et al. (2019). This is due to unavoidable technical issues which prevented the re-use of the Cook 2019 classifier on the different machine architecture and setup used to process the imagery for this manuscript. Thus, Appendix B is required and cannot be removed.

*The surface type classification section is a critical part for analyzing the changes of algae and weather crust (I guess the authors are trying to focus on these two factors). However, why don't the authors include weathering crust as a surface type when using the random forest method to classify the UAS image and Sentinel-2 image. Again, the authors directly used the surface type classification results from Cook et al. (2019) which didn't consider weathering crust.*

Weathering crust is not a distinct surface type in its own right. As we discuss in detail in this manuscript the weathering crust can range from non-existent to substantial, so it would be near impossible to ascribe constant optical properties to a class labelled "weathering crust". That said, the "clean-ice" category is equivalent to ice without visible light absorbing particles and a well-developed weathered crust. As discussed in this manuscript and in Cook et al. (2019) there are many interlinked feedbacks between weathering crust development and particle accumulation that would necessarily cause ambiguity in the labelling between our $H_{bio}$, $L_{bio}$ and a hypothetical "weathering crust" class.

*Besides, it is not appropriate to make statements based on the results from another under-review paper (Cook et al. 2019).*

The only alternative would have been to delay submission of this manuscript pending a final decision on Cook et al. (2019). The review of Cook et al. (2019) has taken significantly longer than anticipated. Both authors have now come to the end of their short-term research contracts and have moved onto other employment, so delaying further was unviable. Finally, we could understand these concerns if we were relying on a paper undergoing closed peer review, but Cook et al. (2019) is undergoing public discussion.

*What's the criteria to separate the high algae surface from low algae surface? Using thresholds? How to define the threshold? The high vs low algae sound very arbitrary.*

As outlined at P5 L19-20, the measurements were initially labelled/separated into low versus high algae by visual examination of the ice surface at the time of spectra acquisition and biological sampling. Subsequent laboratory analysis shows that the biomass concentration of low versus high sites differs by at least an order of magnitude (P5 L21-23) and there is a statistically significant difference between the albedo of the surfaces in each class (see Cook et al. 2019). There is therefore a clear empirical justification for the class definitions.

There remain significant barriers to quantifying glacier-algal biomass on a continuous scale from remote sensing data. Many of these are explained in our manuscript and relate to the highly variable optics of the ice upon which the light absorbing particles rest. The underlying ice-albedo can vary by tens of percent independently of the impurities present on the surface and this can dampen or exaggerate spectral features that could otherwise scale with algal abundance. Therefore, without deep knowledge of the optics of the underlying ice and its spatiotemporal variability, extracting cell abundance is not achievable. This manuscript represents a stride towards gaining that knowledge, but there remain major gaps in the

available empirical data and theory preventing remote biomass quantification over ablating ice. For these reasons, discrete classification remains the best option for quantifying algal coverage and has enabled us to quantify spatio-temporal changes in algal coverage.

*As the authors mentioned that the remote sensing data they were using don't contain spectral signature of algae, in this case, how could the algae surfaces be classified? In other words, how to separate them from other impurities?*

There is a difference between discrete diagnostic features and separable spectra. While previous efforts to remotely sense biological particles in the cryosphere have relied upon discrete features (e.g. Painter et al.'s (2001) 680 nm absorption feature, Takeuchi et al.'s (2006) SPOT carotenoid feature and the vegetation red-edge) they all rely upon sufficient spectral resolution in specific parts of the solar spectrum and also on sufficient knowledge of the background optics. Biomass quantification in these cases also relied upon an assumption of constant spectral albedo in the underlying ice, a lack of confounding inorganic particles and a sufficiently high signal-noise ratio to overcome spectral mixing in heterogeneous pixels.

Our manuscript shows that for ablating ice on the GrIS we cannot assume these conditions are satisfied. Instead, we have used a classifier that uses information from multiple bands ranging from the short visible to the red/near-infrared to separate sample surfaces into classes based upon a training set comprising end-member spectra. In previous work we have demonstrated using spectrometry geochemical analysis and radiative transfer modelling that mineral dusts are not confounding the classification scheme (Cook et al. 2019). We consider this to be more robust to the variable environmental conditions and cross-sensor issues encountered in remote sensing over the ablating GrIS. We also highlight that we have been exceptionally upfront about the limitations of our approach – indeed these issues form some of the central themes of the paper. We point to our section 4.4 in particular.

*Page 11 line 4-15, this part is very unclear, rephrasing is necessary.*

We respectfully note that the topics covered in this section are complex and with a high degree of inter-connectivity. They would benefit from being carefully re-read as we offer a range of possible explanations for the surface changes that we observed. The present text in this section is the result of several drafts. We would welcome specific critique on the issues which are unclear but otherwise we have no suggested changes to this section.

*Although the authors emphasized the importance of weathering crust on surface albedo, but I didn't find detailed quantitative analysis about this subject? Section 4.3 reads quite speculative. Any references to use 840nm to identify the weathering crust?*

There is very little quantitative analysis of weathering crusts in the literature, let alone their specific importance to surface albedo. We believe that our study is the first to explicitly

highlight and examine the importance of weathering crust status upon bare ice albedo variability. We found no studies explicitly about the use of 840 nm as an indicator of weathering crust status, and so we rely on analysis of spectra and glacier algae that has been published previously to deduce that 840 nm is an appropriate indicator of weathering crust status within our study area. We would welcome pointers in the direction of any literature that directly supports use of 840 nm as a weathering crust indicator. One of the main contributions of this manuscript is the first remote mapping of weathering crust properties and the identification of the 840 nm band as an indicator is an important part of that.

*The authors aimed to analyze the impact of algae and weathering crust on Greenland ice sheet albedo, but the datasets are limited to two specific sites. Discussion about the generalization of those two specific sites to larger spatial scale is necessary.*

This is an intrinsic problem of field measurements, which are necessarily limited in spatial extent. We are extremely hesitant in making any grander claims relevant to larger spatial scales without hugely increasing the scope of the remote sensing element of the manuscript, which is not the primary focus of the study: rather, the highly detailed UAS measurements (time series at S6; first observations undertaken at UPE) constitute the key novelty of this study.

*Regarding the scale problem, particularly the impact of scale on melt flux estimation (page 15 line 11-18), did the authors use the actual MODIS albedo to estimate the melt flux? Please clarify. It seems that all the melt flux estimates are based on Sentinel-2 albedo, one scenario is to use Sentinel-2 albedo for each individual Sentinel-2 pixel, another scenario is to calculate an average Sentinel-2 albedo over a MODIS pixel scale. I don't think this comparison is fair, what the difference between the Sentinel-2 averaged albedo and the real MODIS albedo? Without considering this, "the _2% underestimate in melting derived from surface energy budget calculations which use only MOD10A1 albedo" (in abstract and conclusion) is wrong.*

We respectfully disagree that our conclusions are "wrong", but we will improve our wording to avoid confusion. Fundamentally, the section to which R3 refers is a sensitivity analysis which aims to understand the importance of the spatial scale of albedo measurements employed to calculate surface melt volumes. It is therefore not a direct comparison of Sentinel-2 to MODIS, but rather an illustration of the under-estimate in melt volume which can result from using coarse-resolution (500 m) albedo measurements rather than medium-resolution (~20 m) measurements. As we have shown, this is because, at 500 m resolution, the sub-pixel albedo distribution becomes distinctly non-normal and left-skewed. To be clear: this section does not attempt to analyse the absolute differences in measured albedo between S2 and MODIS.

**We therefore propose modifying the text in the relevant locations as follows: "**the ~2% underestimate in melting derived from SEB calculations which only **use albedo measurements at coarse scales such as those in the 500 m MOD10A1 product".**

We will also make the following change to P15 L13: 'We then applied these melt rates to each S-2 pixel **as a function of their albedo value**'.

*Besides, using only two MODIS pixel to make this statement is not sound. The authors should consider conducting the calculation over a large scale, since Sentinel-2 image and MODIS image can cover a large area instead of two MODIS pixels.*

Within the scope of our study, which is tightly focused upon two locations on the west GrIS, we consider that using only the two pixels which directly intersect our study site for which we could collect temporal data is a reasonable methodological approach. From our UAS measurements we are confident that most, if not all, of the Sentinel-2 pixels intersecting with the MODIS pixels have relatively uni-modal and un-skewed distributions. If applied to other MODIS pixels outside the study area then this would have to be assumed as we do not have the UAS measurements to back this up. As such, it is unclear to us how conducting these proposed calculations over a large scale would improve the conclusions that we reach. We also refer to our response to a previous comment, that this exercise was not intended as a MOD10A1 validation exercise per-se but instead as a study of the impact the coarsening spatial resolution of albedo products makes upon melt rate calculations.

---

## Author Response (AR2)

**Response to review of TC-2019-131 v.2**

*Tedstone et al. use field observations to demonstrate that 1) pigmented glacier algae are ubiquitous across western Greenland bare ice, 2) algae reduce albedo both within and outside the dark zone, 3) surface topography impacts algal distribution and therefore is important for albedo variability, 4) weathering crust development is important for determining albedo variability, and 5) there is a spatial scale dependency on albedo measurement which impacts detection of real albedo changes at coarse resolutions. I find the albedo analysis (i.e. points 1, 2, 3 and 5) to be sound. However, after reviewing this revised version, and considering some of the other reviewer comments, I am not convinced about point 4. As such, I cannot endorse this manuscript without some changes. In my opinion, the authors should focus on albedo and its scale dependencies which are the most robust parts of the manuscript.*

We thank the reviewer for their appraisal that they find points 1, 2, 3 and 5 to be sound. But more importantly, we thank the reviewer for their critique of point 4, which caused us to investigate our primary data sources in more detail. We believe that the additions we have made to the manuscript are a significant improvement. We introduce these below.

*Major comment*
*According to the authors, "One of the main contributions of this manuscript is the first remote mapping of weathering crust properties and the identification of the 840 nm band as an indicator is an important part of that." However I remain skeptical that the weathering crust "status" can truly be determined by reflectance in the 840 nm band. This point was also raised by Reviewer 3.*

*First of all, it is not clear what the 840 nm band is responding to. In the manuscript, the authors state that they "believe that the dominant signal retrieved at 840 nm by our UAS is indicative of the weathering crust state, inclusive of ice grain sizes, ice density, porosity and interstitial and surface meltwater." However, the authors present no evidence (either theoretically, conceptually or observationally) that this is true. For example, the increase in 840 nm reflectance between July 20 and 21 (Fig. 3) could just as likely be due to decreasing interstitial water content after the rainfall event. This would require no change in "weathering crust structure" (or that it "drives albedo variability" as the title implies). Without isolating the mechanism causing the changes in the 840 nm band the authors cannot really argue that the weathering crust causes variability in western Greenland Ice Sheet albedo. In which case, the title and focus of the manuscript should really change. This would likely require some structural revisions because the manuscript currently makes it sound like the authors quantitatively proved that weathering crust structure drives albedo variability.*

We agree that some aspects of our previous evidence base were not sufficiently well-explained. The key point we have been aiming to highlight with this part of our study is that ice surface properties and albedo can change significantly from one day to the next for reasons unrelated to surface impurity loading, which is demonstrable despite there still being outstanding questions to resolve in future studies. Current understanding of the co-evolution of glacier ice weathering crust and albedo is still in its infancy.

In the revised manuscript we provide new evidence to illustrate changes in weathering crust state: (a) measurements of surface lowering for comparison against modelled estimates of

surface melting, (b) a series of oblique photographs of the ice surface in the UAS area taken during the period of interest, and (c) closer references to the small literature base which concerns weathering crust processes.

In summary, the new observations which we present clearly show that weathering crust collapse occurred during storm conditions on 18-19 July 2017. We observed 2.5 cm of snow melt and then > 9 cm of ice surface lowering within a 30-hour period, which is at least twice the amount of water-equivalent melting predicted by our SEB model. Weathering crust collapse, leaving areas of denser ice, often with water ponded on top, are clearly visible in the accompanying photographs. Please see the revised manuscript for full details.

The reviewer was concerned that *'the increase in 840 nm reflectance could….be due to decreasing interstitial water content….[requiring] no change in weathering crust structure'*. We concur and so have modified our wording to discuss WC-driven albedo changes in terms of the overall 'state' of the weathering crust, i.e. including interstitial and perched meltwater, rather than just the matrix structure. In particular, we have modified the manuscript's title to 'Algal growth and weathering crust state…' (from structure) and now use 'state' in the manuscript body.  We wish to reinforce the point here that, as far as albedo is concerned, the key issue seems to be the presence or absence of air-ice interfaces (after Jonsell et al 2003), rather than the weathering crust matrix structure itself. Clearly, a well-developed WC matrix that is filled with water will have few air-ice interfaces, so its light scattering potential is likely more similar to denser ice. Since the refractive indices of frozen and liquid water are near identical, disentangling meltwater accumulation from ice density is a difficult problem. In summary, we hope that the shift away from claiming to quantify WC *structure* will resolve these concerns.

*If the authors could demonstrate that 840 nm reflectance is responding to weathering crust thickness/porosity then the authors should be able to identify weathering crust these changes in their data and add more classes to their supervised classification. Regardless of it being a continuum, the authors could identify categories (i.e. extensive to non-existent) in the same way as they did for algal concentration. This analysis would allow the authors to quantify the relative importance of weathering crust versus algae and complete one of the main goals of this manuscript.*

The problem with adding classes to the supervised classification is that it would increase ambiguity. We already show that even with a simple 'clean ice'-'light algae'-'heavy algae' split, there is already significant uncertainty in whether changes in the class of individual pixels from one day to the next are actually driven by glacier algae population changes or other processes such as weathering crust development.

The underlying problem here is that this approach would necessitate a highly multi-way classifier which we have no means of training using our small (171-member) training dataset. We would first need to identify some way of labelling each member with weathering crust status, but this is not realistically possible as these labels were not defined while taking the field measurements. Even then, assuming this issue could be resolved, introducing 'no crust', 'medium crust', 'thick crust' would yield six further categories overall, with a commensurate reduction in the number of training samples available for each category and hence significant reductions in classifier recall and accuracy.

In light of these ambiguities and limitations we are not prepared to implement a classifier with additional categories. It would weaken, rather than strengthen our conclusions.

*Other than this major comment, I found that several clarifications and improvements to the grammar are still required. These are detailed in my comments below. Please note that my page and line numbers refer to the tracked changes version of the manuscript.*

*Specific comments*

*P1 L1: Consider clarifying why albedo matters for a non-specialist reader.*

Introduction sentence now reads:

**One of the primary controls upon the melting of the Greenland Ice Sheet (GrIS) is albedo, a measure of how much solar radiation that hits a surface is reflected without being absorbed. Lower-albedo snow and ice surfaces therefore warm more quickly.**

*P1 L3: Check for consistency, "bare ice" or "bare-ice"?*

**Done**

*P1 L7-8: What is the difference between "underlying ice properties" and "weather crust development"?*

Changed to **'…including modification of the ice surface by algal bloom presence through locally enhanced melting…'.**

*P1 L9: missing an "an" between "use" and "unmanned".*

No 'an' is needed as we referred to 'observations', but we have still **modified this sentence** to improve readability.

*P2 L4: Consider replacing "and" with "so"*

**Done.**

*P2 L8-9: van den Broeke et al. (2017) is a review paper. Suggest crediting the study that observed this albedo change.*

Whilst van den Broeke et al. (2017) is indeed a review paper, their Figure 4 upon which our reference is based has not been published elsewhere previously. Their manuscript references to albedo decline are to Box et al. (2016, GEUS Bulletin) and Tedesco et al. (2016), do not extend as far as 2017, unlike their Fig. 4. Our reference is therefore correct.

*P4 L28-29: How do you know that bare ice surfaces are non-Lambertian? Suggest adding a reference.*

**We have added a reference to Knap & Reijmer (1998).** Nevertheless, It is also contrary to established theory on light scattering by snow and ice to suggest that they are Lambertian scatterers. Where bare ice surface are rough and weathered they are *closer* to Lambertian because of multiple scattering in the ice volume. Smoother, denser ice is further from Lambertian because of a specular reflection peak in the forward hemisphere. Not even dry snow is Lambertian. We are not aware of any natural surface that is truly Lambertian.

Knap & Reijmer also showed that narrowband-to-broadband retrievals from Landsat TM2 versus TM4 result in different over- vs under-estimation tendencies, so we have removed the statement that albedo retrievals based on nadir measurements are always under-estimates.

*P6 L23: Suggest clarifying the method which found glacier algae to be ubiquitous. UAV remote sensing or biological sampling?*

Added **'Biological sampling and UAS observations showed that…'**

*P7 L6-8: These sentences would be better moved the results of the surface classification section.*

Thanks. **Moved as suggested.**

*P8 L8-11: This is speculation. Either present evidence for this or clarify that this is your hypothesis.*

As we already note, some of our team were present at the site during June and observed this process (see Section 2 'Study sites'). We have bolstered our argument by providing an oblique photograph illustrating local high concentrations of flushed impurities at a change-in-gradient of a supraglacial stream (Supplementary Figure D2). The sentence in question now reads as follows:

> **However, during June, winter snowpack retreat caused significant ephemeral water drainage pathways to develop, causing algal cell re-distribution (e.g. Suppl. Fig. D2).**

*P9 L15: Clarify why a weathering crust increases scattering opportunities. More ice-air interfaces?*

This section has been re-structured in response to the major comment, please see new text.

*P9 L16: What do you mean by "weathering crust status"? Thickness? Porosity?*

Later in the paragraph we define this term to be 'inclusive of ice grain sizes, ice density, porosity and interstitial and ponded surface meltwater' (previous m/s P9 L21-22). **Note that we have changed from 'status' to 'state' for this revision.**

*P9 L16-17: Some more evidence is required here to convince the reader that absorption in 840 nm is due to weathering crust properties. See major comment.*

We believe that this issue has been dealt with through our inclusion of additional observations and discussion that we introduce above.

*P16 L10: Consider backing up this statement.*

Added spatial qualification: **"Glacier algae are ubiquitous in the two areas of the western GrIS ablation zone that we surveyed."**

*P16 L20-21: There are no "physical weathering crust changes" presented in this manuscript. Suggest adding a reference here so that it does seem like a finding of the study.*

We now present additional evidence that prove that we observed physical WC changes.

*P16 L21-23: What do you mean by "state" of weathering crust? Water content? Thickness? Porosity?*

This refers to the earlier definition (P9, L21-22 in previous version) but we have also added **"(i.e. density/porosity, interstital and ponded water content)"**.

*P16 L25: Sentinel-2 or S-2? Be consistent.*

For the conclusion, Sentinel-2. There is always the possibility that a reader skips the body of the text and so we prefer to re-state the full satellite name in this case. We have no problem with this being changed in type-setting if editorial guidelines require otherwise.

[revised manuscript text omitted]